# Construction and Identification of Cold Tolerance in Different Broccoli Cultivars at the Seedling Stage

Dongna Wen [1], Fengqing Han [1], Yongyu Zhao [1], Yuxiang Liu [1,2], Yumei Liu [1], Jianxin Huang [1] and Zhansheng Li [1,*]

[1] State Key Laboratory of Vegetable Biobreeding, Institute of Vegetables and Flowers, Chinese Academy of Agricultural Sciences, Beijing 100081, China; wendongna0713@163.com (D.W.); hanfengqing@caas.cn (F.H.); zhaoyongyu0212@163.com (Y.Z.); liuyuxiang822@163.com (Y.L.); liuyumei@caas.cn (Y.L.); hjx5168@163.com (J.H.)

[2] Key Laboratory for Vegetable Biology of Hunan Province, Engineering Research Center for Horticultural Crop Germplasm Creation and New Cultivar Breeding, Ministry of Education, Hunan Agricultural University, Changsha 410128, China

* Correspondence: lizhansheng@caas.cn

**Abstract:** This study aimed to develop a systematic method for assessing the cold tolerance of broccoli seedlings, which is helpful to evaluating the cold tolerance of broccoli cultivars. We selected 14 classical broccoli cultivars and evaluated their cold tolerance by examining physiological indicators including the levels of cold resistance, electrical conductivity, malondialdehyde (MDA), soluble sugar, and hydrogen peroxide ($H_2O_2$). The results showed that there were significant differences in detected characteristics in all cultivars after cold stress during the seedling stage ($p < 0.05$). Among them, Meiqing had the strongest cold tolerance, followed by King 11, Zhongqing 15, Meiao 7172, Zhongqing 318, Zheqing 80, Zhongqing 16, Zhongqing 319, and Lvxiong 90. However, Zhongqing 11, Yanxiu, Qianghan, and Feicui 5, showed the worst cold tolerance (all died). Pearson correlation analysis indicated that there was a significantly negative correlation between the cold tolerance and the electrical conductivity during broccoli seedling stage, with a correlation coefficient of −0.586 ($p < 0.05$). At the same time, we found that the electrical conductivity of all broccoli cultivars showed a positive correlation with the MDA and soluble sugar levels, with the correlation coefficients 0.650 and 0.573, respectively ($p < 0.05$). This study not only firstly provides a fundamental method for evaluating the cold tolerance in different broccoli genotypes and the other cruciferous vegetables, but also offers a scientific evidence explaining the cold tolerance of the Meiqing, King 11, and Zhongqing 15 cultivars widely cultivated in China.

**Keywords:** broccoli; cultivar; cold resistance; genotype; electrical conductivity

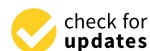



## 1. Introduction

Broccoli (*Brassica oleracea* L. var. italica) is a nutritious Brassica crop. Its edible flower head is packed with vitamins, fiber, phenolic compounds, and glucosinolates. These natural substances offer anticancer, antioxidant, and cardiovascular benefits [1–3]. Broccoli holds significant economic and nutritional value and enjoys global popularity as a vegetable.

However, the frequent occurrence of extreme weather, high temperature, drought, cold damage, freezing damage, and other abiotic stresses seriously affect the harvest of broccoli [4]. Temperature directly affects the growth process, final yield, and flower quality of broccoli. Under low-temperature stress, the growth points of broccoli seedlings freeze and deteriorate, resulting in slow growth, reduced yield, and no harvest. Broccoli bulbs turn purple at low temperatures, losing their commercial value and causing large economic losses [5]. In this context, it is particularly important to screen cold-resistant cultivars of broccoli, and cold resistance identification has emerged. Previous studies have shown that under low-temperature stress, the content of reactive oxygen species in plants increases and reacts with unsaturated fatty acids on the plasma membrane to produce MDA, leading

to membrane lipid phase transition, increased cell membrane permeability, and increased electrolyte permeability. At the same time, low temperature also affects the synthesis of chlorophyll, leading to a decrease in the photosynthetic rate and changes in the content of soluble sugars, $H_2O_2$, and other substances in plants [6–9]. Most of the existing research on low temperature in broccoli has focused on vernalization and the impact of exogenous factors on broccoli plants, such as the effects of exogenous $Ca^{2+}$ and ABA on broccoli under low-temperature stress [10–12]. There is very little research on the comprehensive identification of low-temperature tolerance, and plant low-temperature tolerance is a comprehensive trait that requires the use of a combination of multiple methods to avoid the one-sided evaluation of plant low-temperature tolerance using only a single indicator of change. To date, only a small number of researchers have carried out a comprehensive evaluation of low-temperature tolerance on several broccoli materials and screened broccoli cultivars with notable low-temperature tolerance by determining and analyzing the MDA content, electrolyte osmolality, soluble protein content, and chlorophyll content of broccoli leaves under low-temperature stress, but the purpose of these studies to screen out broccoli cultivars with stronger cold tolerance and to provide a material basis for cultivating new cold-tolerant germplasm in broccoli.

The above methods for cold tolerance identification have drawbacks such as poor stability, long cycles, and high detection difficulty. Currently, there is a lack of an efficient, convenient, and systematic rapid cold tolerance identification method. It is urgent to establish a fast and reliable detection technology system for screening and identification of broccoli resources, gene mining, and analysis, etc.

Late-maturing broccoli varieties typically possess cold tolerance, while early-maturing broccoli varieties typically lack cold tolerance. Existing studies have shown that low temperatures can induce the accumulation of anthocyanins in most plant bodies [13]. However, considering the complexity of anthocyanins, we did not use them as measurement indicators. This study measured five important physiological indicators of the electrical conductivity, MDA, soluble sugar, $H_2O_2$ content, and chlorophyll, in the leaves of 14 main cultivated broccoli cultivars in China after low-temperature treatment at the seedling stage. By analyzing the correlation between phenotypic traits and physiological index data, a fast and stable harmless identification system for cold resistance of broccoli seedlings was constructed, and nondestructive detection indices were determined to better quantify and evaluate the cold resistance of different cultivars of broccoli, provide a basis for screening cold-tolerant cultivars of broccoli, and lay a foundation for the subsequent improvement of cultivars.

## 2. Materials and Methods

### 2.1. Plants, Instruments, and Reagents

Fourteen broccoli cultivars currently grown in China, were collected and tested in this study (Table 1). All the materials numbered B1–B14 were sown on 29 December 2022, and the seedling with four leaves were treated for tolerance test on 3 March 2023. All the samples were set three treatments (n = 3) and every treatment included 5 plants.

A perforator (Beijing Rongxing Bio-Technology Co., Ltd., Beijing, China), visible spectrophotometer (Beijing Ruixinuo Bio-technology Co., Ltd., Beijing, China), water bath (Shanghai Changshuo Bio-technology Co., Ltd., Shanghai, China), table centrifuge (Beijing Hua Yue Yang Bio-technology Co., Ltd., Beijing, China), pipette guns (Beijing Yishan Huitong Technology Co., Ltd., Beijing, China), Olympus BX51 microscope (Olympus Corporation, Tokyo, Japan). The slides, coverslips, tweezers, mortars, and 96-well plates used in this study were sourced from Beijing Zhongchuang Hongda Technology Co., Ltd., Beijing, China. The anhydrous ethanol, distilled water, acetone, and concentrated hydrochloric acid used in this study were sourced from Sinopharm Chemical Reagent Co., Ltd., Shanghai, China. And some kits including BC0025, BC0035, and BC3595 kits were obtained from Solarbio Technology Co., Ltd., Beijing, China.

**Table 1.** Information of 14 different broccoli cultivars cultivated in China.

| Number | Cultivar | Generation | Source | Maturity |
|--------|----------|------------|--------|----------|
| B1 | Naihanyouxiu | $F_1$ | Sakada, Japan | Medium early |
| B2 | Yan Xiu | $F_1$ | Sakada, Japan | Medium to late |
| B3 | Qianghan | $F_1$ | Saint Denis, USA | Medium to late |
| B4 | Jade No. 5 | $F_1$ | Syngenta | Medium |
| B5 | Lvxiong 90 | $F_1$ | Shida, Japan | Late |
| B6 | King 11 | $F_1$ | Zhaofeng, Wenzhou | Extremely late |
| B7 | Zheqing 80 | $F_1$ | Zhejiang Meizhiao Seed Industry | Extremely late |
| B8 | Zhongqing 11 | $F_1$ | Chinese Academy of Agricultural Sciences | Extremely precocious |
| B9 | Zhongqing 15 | $F_1$ | Chinese Academy of Agricultural Sciences | Medium to late |
| B10 | Zhongqing 16 | $F_1$ | Chinese Academy of Agricultural Sciences | Precocity |
| B11 | Zhongqing 318 | $F_1$ | Chinese Academy of Agricultural Sciences | Extremely late |
| B12 | Zhongqing 319 | $F_1$ | Chinese Academy of Agricultural Sciences | Late |
| B13 | Meiqing | $F_1$ | Zhejiang Meizhiao Seed Industry | Medium to late |
| B14 | Meiao 7172 | $F_1$ | Zhejiang Meizhiao Seed Industry | Medium to late |

*2.2. Cold-Resistant Test and Phenotypic Investigation*

All broccoli materials were used as controls before cold-resistant treatment. Broccoli seedlings were first treated in a low-temperature incubator at 4 °C for 2 days, and then at 2 °C/–5 °C (16 h/8 h) for 3 days. After cold-resistant treatment, all broccoli samples were transferred to room temperature (25 °C) and cultured for 3 days. According to the degree of damage to the leaves of broccoli seedlings and the survival rate of the seedlings, their cold resistance can be divided into the following six levels, as shown in Table 2.

**Table 2.** Broccoli cold resistance grade classification standards.

| Cold Resistance Rating (0–5) | Performance | Cold Resistance |
|------------------------------|-------------|-----------------|
| 0<br>1 | There is little or only slight damage to the leaves of broccoli seedlings. Most leaves have no obvious yellowing, wilting, or necrosis. | Highly cold tolerant |
| 2<br>3 | The leaves of broccoli seedlings are damaged to a certain extent, and some leaves are slightly yellowed, wilted, or edge browned. | Moderate cold tolerance |
| 4<br>5 | The leaves of broccoli seedlings were severely damaged, and most of the leaves showed obvious yellowing, wilting, browning, or necrosis. | Low cold tolerance |

*2.3. Determination of Electrical Conductivity*

Fresh broccoli leaves were collected for electrical conductivity value test. Firstly, using a punch to take small leaves with a diameter of 1 cm, and weighting 0.2 g samples in each test tube, adding 10 mL distilled water, setting up a control group just with 10 mL distilled water. Holding 12 h at room temperature, the electrical conductivity of the test group was recorded as $E_1$, and then boiled for 35 min. After cooling, the electrical conductivity was recorded as $E_2$, and the control was recorded as $E_0$. The electrical conductivity was evaluated by the following formula: $E = (E_1 - E_0)/(E_2 - E_0) \times 100\%$ [14].

### 2.4. Determination of MDA

The supernatant of all the sample to be tested was obtained according to the MDA content detection instruction bc0025, and 200 μL was added to a 96-well plate. The absorbance of each sample was measured at 532 nm and 600 nm, and $\Delta A532 = \Delta A532_{determination} - \Delta A532_{blank}$, $\Delta A600 = \Delta A600_{determination} - \Delta A600_{blank}$, $\Delta A = \Delta A532 - \Delta A600$ (only 1–2 times for blank tube). Then, the MDA level was calculated with the following formula: MDA content (nmol·g$^{-1}$ mass) = [$\Delta A \times$ V reverse total ÷ ($\varepsilon \times$ D) $\times$ 109] ÷ (W $\times$ V sample ÷ V extraction) $\times$ F = 53.763 $\times \Delta A$ ÷ W. V is the total volume of the reaction system ($5 \times 10^{-4}$ L), $\varepsilon$ is the molar absorptivity of MDA ($1.55 \times 10^5$ L), V is the volume of added sample (0.1 mL), D is the optical diameter of the 96-well plate (0.6 cm), V is the volume of added extract (1 mL), W is the sample mass, and F is the dilution factor [15].

### 2.5. Determination of the Soluble Sugar Level

The sample solution to be tested was obtained according to the soluble sugar content detection instruction BC0035, and 200 μL was added to a 96-well plate. The absorbance was measured at 620 nm and recorded as A blank tube, A measuring tube, and A standard tube in the following equation: $\Delta A = A_{measuring\ tube} - A_{blank\ tube}$, $\Delta A_{standard} = A_{standard\ tube} - A_{blank\ tube}$ (only 1–2 tubes for the blank tube and standard curve), according to the concentration (y, mg·mL$^{-1}$) and absorbance of the standard tube $\Delta A$ standard (x, $\Delta A$ standard). A standard curve was established. According to the standard curve, $\Delta A$ (x, $\Delta A$) was entered into the formula to calculate the sample concentration (y, mg/mL), soluble sugar (mg·g$^{-1}$ mass) = (y $\times$ V1)/(W $\times$ V1/v2) = 10 $\times$ y/W, where V1 is the added sample volume (0.04 mL), V2 is the total sample volume (10 mL), and W is the sample mass [16].

### 2.6. Determination of $H_2O_2$ Content

The sample solution to be tested was obtained according to the $H_2O_2$ content detection instructions for the BC3595 kit, and 200 μL was added to a 96-well plate. The absorbance value was measured at 415 nm and recorded as A blank tube, A measuring tube, and A standard tube in the following equation: $\Delta A = A_{measuring\ tube} - A_{blank\ tube}$ $\Delta A_{standard} = A_{standard\ tube} - A_{blank\ tube}$ (only 1–2 tubes for blank tube and standard curve). $H_2O_2$ content (μmol·g$^{-1}$ mass) was determined as follows: $\Delta A$ determination/($\Delta A$ standard/C standard solution) $\times$ V sample/(V sample/V extraction $\times$ W) = 2 $\times \Delta A$ determination/$\Delta A$ standard/W, where C standard solution is the concentration of $H_2O_2$ standard solution (2 μmol·mL$^{-1}$), V sample is the volume of added sample (0.25 mL), W is the tissue mass, and V extraction is the volume used in the extraction process (1 mL) [17].

### 2.7. Determination of Chlorophyll Content

Select leaves from the same part of broccoli seedlings. Weigh 0.1 g of fresh sample into a mortar, add 1 mL of distilled water and a small amount of calcium carbonate (approximately 10 mg), thoroughly grind under dark or dim light conditions, and transfer to a 10 mL test tube. Rinse the mortar with the extraction solution prepared in a volume ratio of anhydrous ethanol to acetone of 1:2. Transfer all the rinsing solution to a 10 mL test tube and make up to 10 mL with the extraction solution. Place the test tube in darkness or cover it with aluminum foil for 3 h of extraction. The absorption value was measured at 663 nm and 645 nm using a wavelength detector and denoted as $A_{663}$ and $A_{645}$, respectively. Total chlorophyll concentration was evaluated by the following formula: $C_T$ (mg·g$^{-1}$) = 8.02 $A_{663}$ + 20.21 $A_{645}$ [18].

### 2.8. Statistical Analysis

All data shown were mean $\pm$ SD (n = 3) using SPSS Statistics version 19.0 (SPSS, Inc., Chicago, IL, USA). One-way ANOVA and Tukey's multiple-range test were used to evaluate significant differences ($p < 0.05$). A two-tailed Pearson's correlation coefficient analysis and linear regression analysis were performed to evaluate the cold tolerance of fourteen broccoli cultivars and relevance of the electrical conductivity, MDA, soluble

sugar, and $H_2O_2$ physiological indicators ($p < 0.05$). The statistical analyses and charts were produced using the IBM SPSS and GraphPad Prism 8.2.0 (GraphPad Software Inc., San Diego, CA, USA) software packages, respectively.

*2.9. Stomatal Observation*

The leaves of 14 species of broccoli were sampled, suitable leaves were selected and cut, the lower epidermis of the leaves was taken with forceps, the obtained lower epidermis was placed on a slide with a drop of distilled water, and after the lower epidermis was completely unfolded the coverslip was put on, and the observations were kept in the photographs. The same procedure was repeated after the low-temperature treatment and the photographs were observed.

## 3. Results

*3.1. The Phenotypic Profiles of Broccoli Response to Cold Stress*

From Figure 1 and Table 3, we can see that there were obvious responses of fourteen broccoli cultivars to the cold stress. Among them, the performance of Meiqing to the cold stress was the best with a high cold resistance value of 5, and few plants showed leaves turning yellow. Meanwhile, there was no damage to the heart leaves of broccoli Meiqing (cold resistant). The cultivars of King 11, Zhongqing 15, Meiao 7172, Zhongqing 318, Zheqing 80, Zhongqing 16, Zhongqing 319, and Lvxiong 90, also showed excellent cold tolerance, with the same cold resistance value of 4. However, broccoli cultivars of Zhongqing 11, Yanxiu, Qianghan, and Feicui 5 all showed poor cold tolerance (all died), with the same cold resistance value of 0 and all leaves losing water and yellowing.

**Table 3.** The cold tolerance values of 14 broccoli cultivars.

| Number | Cultivar | Cold Resistance Rating (0–5) |
|--------|----------|------------------------------|
| B1 | Naihanyouxiu | 2 |
| B2 | Yan Xiu | 1 |
| B3 | Qianghan | 1 |
| B4 | Jade No. 5 | 0 |
| B5 | Lvxiong 90 | 3 |
| B6 | King 11 | 4 |
| B7 | Zheqing 80 | 3 |
| B8 | Zhongqing 11 | 2 |
| B9 | Zhongqing 15 | 4 |
| B10 | Zhongqing 16 | 3 |
| B11 | Zhongqing 318 | 4 |
| B12 | Zhongqing 319 | 3 |
| B13 | Meiqing | 5 |
| B14 | Meiao 7172 | 4 |

*3.2. Electrical Conductivity Response to Cold Stress*

The electrical conductivity of different broccoli cultivars showed a large range of changes and significant differences before and after cold stress ($p < 0.05$) (Figure 2). Before low-temperature treatment, the electrical conductivity of B1–B14 broccoli leaves ranged from 9.45% to 16.98%, with an average value of 12.6%. Among them, the highest electrical conductivity was observed in B7 (Zheqing 80), and the lowest was B10 (Zhongqing 16). After low-temperature treatment, the electrical conductivity of broccoli B1–B14 increased and ranged from 23.77% to 90.89%, with an average value of 58.05%. And the highest electrical conductivity was found in B2 (Yanxiu), and the lowest was found in B7 (Zheqing 80). There were different significant differences in the electrical conductivity between B1–B14 before and after low-temperature treatment, except for B7 and B10 (Figure 2a), while there were also significant differences in the electrical conductivity between B1 and B14 after low-temperature treatment (Figure 2b).

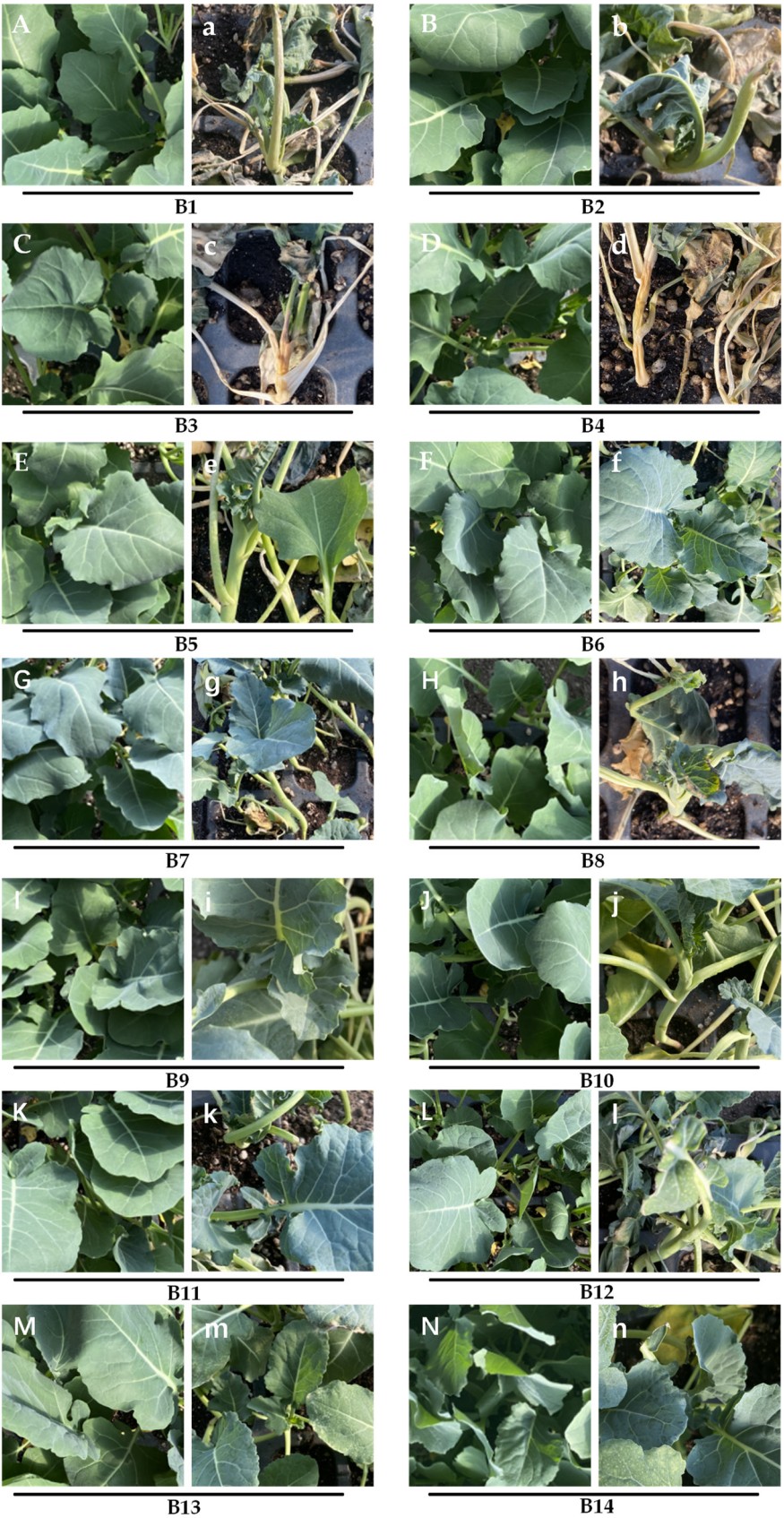

**Figure 1.** Comparison of 14 types of broccoli before and after low-temperature treatment. Note: The letters (**A**(**a**)–**N**(**n**)), respectively, represent B1 to B14, and the capital letters represent the profiles of plants before and after low-temperature treatment.

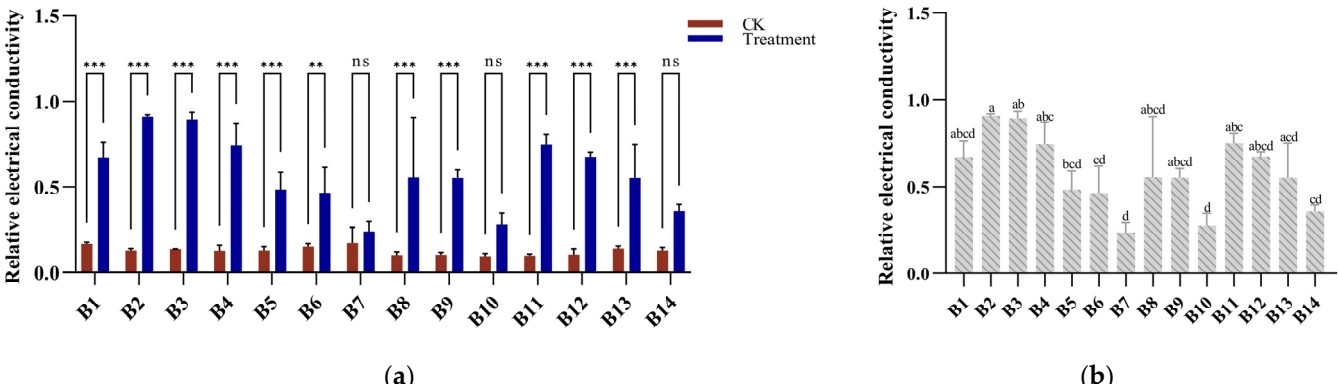

(**a**)  (**b**)

**Figure 2.** Responses of electrical conductivity of different broccoli cultivars to cold stress. (**a**) The comparisons of the electrical conductivity among broccoli cultivars before and after low-temperature treatment. CK refers to before low-temperature treatment. Treatment refers to after low-temperature treatment. ** indicates a significant difference at the level of $p < 0.01$, *** indicates a significant difference at the level of $p < 0.001$, ns indicates an insignificant difference. (**b**) The changes in the electrical conductivity in different broccoli after cold stress using one-way ANOVA analysis. Different lowercase letters indicate significant differences at the level of $p < 0.05$.

### 3.3. MDA Response to Cold Stress

The responses of MDA in different broccoli cultivars to cold stress are shown in Figure 3. We found that the MDA levels of B1–B14 broccoli cultivars ranged from 8.423 to 13.98 nmol·g$^{-1}$ FW. Among them, the highest level of MDA was 13. 98 nmol·g$^{-1}$ FW, detected in B13 (Meiqing), while the lowest level was 8.42 nmol·g$^{-1}$ FW, detected in B4. After cold stress, there was a significant difference in MDA contents in broccoli cultivar (Figure 3b), ranging from 10.22 to 28.85 nmol·g$^{-1}$ FW. Among them, B3 was detected with the highest MDA level (28.85 nmol·g$^{-1}$ FW), which was significantly higher than that of B6 and B7. B4 was detected with the lowest MDA level (10.22 nmol·g$^{-1}$ FW), which was significantly lower than that of B3, B2, and B1.

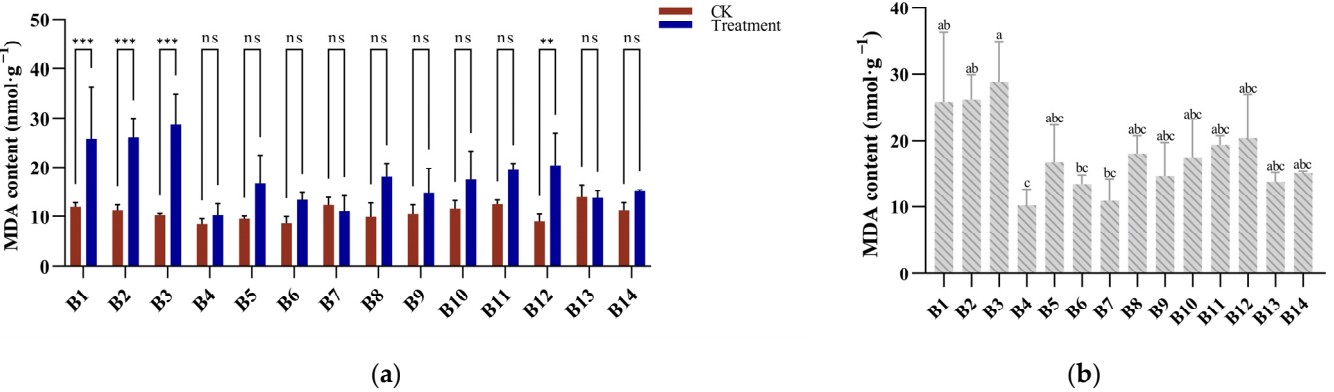

(**a**)  (**b**)

**Figure 3.** Responses of MDA of different broccoli cultivars to cold stress. (**a**) The comparisons of MDA contents among broccoli cultivars before and after low-temperature treatment. CK refers to before low-temperature treatment. Treatment refers to after low-temperature treatment. ** indicates a significant difference at the level of $p < 0.01$, *** indicates a significant difference at the level of $p < 0.001$, and ns indicates an insignificant difference. (**b**) The changes in MDA contents in different broccoli after cold stress using one-way ANOVA analysis. Different lowercase letters indicate significant differences at the level of $p < 0.05$.

### 3.4. Soluble Sugar Response to Cold Stress

As shown in Figure 4a, we could find that the soluble sugar contents of broccoli cultivars ranged from 0.57 to 4.30 $\mu g \cdot g^{-1}$ FW before low-temperature treatment, with an average of 2.19 $\mu g \cdot g^{-1}$ FW. Among them, the highest soluble sugar level was B7 (Zheqing 80), and the lowest was B8 (Zhongqing 11). After cold stress, the soluble sugar content in the leaves of broccoli B1–B14 ranged from 1.21 to 10.70 $\mu g \cdot g^{-1}$ FW, with an average of 4.98 $\mu g \cdot g^{-1}$ FW. And the soluble sugar levels of B1, B2, B3, B4, B8, B9, and B10 increased significantly, while the soluble sugar level of B7 significantly decreased. The highest soluble sugar level was B2 (Yanxiu), which was significantly higher than that in the remaining 10 cultivars except B1, B3, and B9. The lowest soluble sugar level was B7 (Zheqing 80), which was significantly lower than that in B1, B2, B3, and B9. The soluble sugar levels were not significant different among the remaining nine cultivars (Figure 4b).

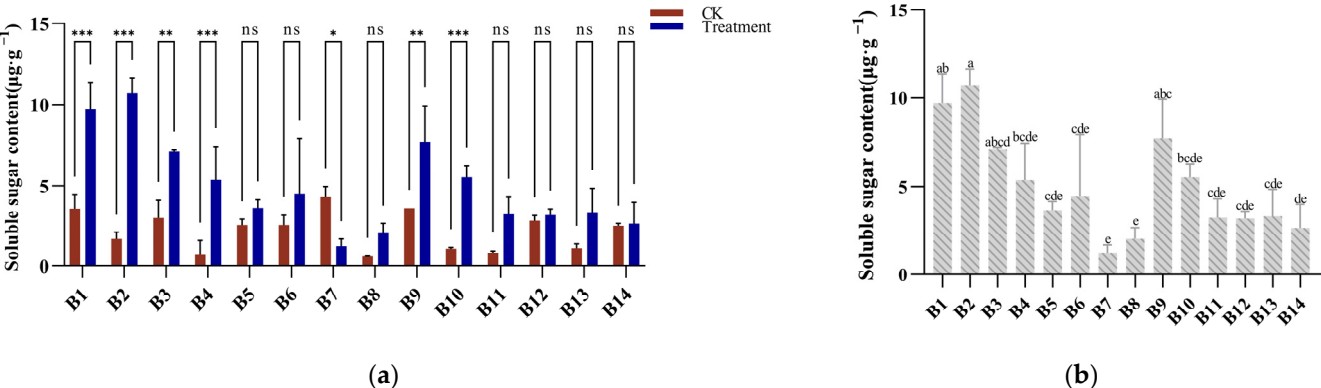

(**a**)            (**b**)

**Figure 4.** Responses of soluble sugar contents of different broccoli cultivars to cold stress. (**a**) The comparisons of soluble sugar contents among broccoli cultivars before and after low-temperature treatment. CK refers to before low-temperature treatment. Treatment refers to after low-temperature treatment. * indicates a significant difference at the level of $p < 0.05$, ** indicates a significant difference at the level of $p < 0.01$, *** indicates a significant difference at the level of $p < 0.001$, ns indicates an insignificant difference. (**b**) The changes in soluble sugar contents in different broccoli after cold stress using one-way ANOVA analysis. Different lowercase letters indicate significant differences at the level of $p < 0.05$.

### 3.5. Hydrogen Peroxide Response to Cold Stress

Before low-temperature treatment, the $H_2O_2$ contents of B1–B14 ranged from 4.49 to 6.84 $\mu mol \cdot g^{-1}$ FW, with an average of 5.44 $\mu mol \cdot g^{-1}$ FW. Broccoli cultivar B8 (Zhongqing 11) was detected with the highest level of $H_2O_2$ while B10 (Zhongqing 16) with the lowest level of $H_2O_2$. After low-temperature treatment, the $H_2O_2$ contents of broccoli cultivars B1–B14 ranged from 3.23 to 8.38 $\mu mol \cdot g^{-1}$ FW, with an average of 5.59 $\mu mol \cdot g^{-1}$ FW. Among them, the highest $H_2O_2$ level was detected in B8 (Zhongqing 11) and the lowest was detected in B7 (Zheqing 80). There were no significant differences in $H_2O_2$ contents in fourteen broccoli cultivars, but significant differences in $H_2O_2$ contents in all the samples of B1–B14 after low-temperature treatment (Figure 5b).

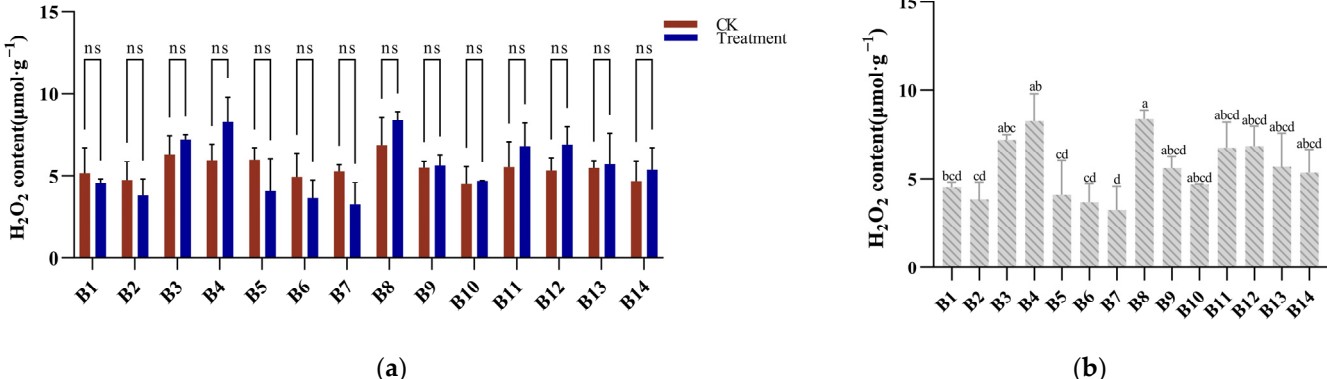

(**a**)                                                                 (**b**)

**Figure 5.** Responses of $H_2O_2$ contents in different broccoli cultivars to cold stress. (**a**) The comparisons of $H_2O_2$ levels among broccoli cultivars before and after low-temperature treatment. CK refers to before low-temperature treatment. Treatment refers to after low-temperature treatment, ns indicates an insignificant difference. (**b**) The changes in $H_2O_2$ levels in different broccoli after cold stress using one-way ANOVA analysis. Different lowercase letters indicate significant differences at the level of $p < 0.05$.

*3.6. Chlorophyll Response to Cold Stress*

From Figure 6a, we could find that the variation range of the chlorophyll contents in fourteen broccoli cultivars was 1.49–2.43 mg·g$^{-1}$ FW, with an average of 1.90 mg·g$^{-1}$ FW before low-temperature treatment. Among them, the highest chlorophyll content was detected in B1 (excellent cold resistance), and the lowest level was detected in B10 (Zhongqing 16). There were different significant differences in the chlorophyll contents in B2, B4, B5, B6, B12, and B13 before and after low-temperature treatment. According comparison, broccoli cultivars of B2, B6, and B13 showed an obvious decrease in chlorophyll content compared to the levels before cold stress treatment. After low-temperature treatment, the chlorophyll contents of broccoli cultivars B1–B14 ranged from 1.52 to 3.32 mg·g$^{-1}$ FW, with an average of 1.99 mg·g$^{-1}$ FW. Among them, the highest chlorophyll content was found in B5 (Lvxiong 90), which was significantly higher than that in the remaining 13 cultivars. The lowest chlorophyll content was found in B8 (Zhongqing 11), followed by B6 (King 11) and B10 (Zhongqing 16). But the chlorophyll contents of the 3 cultivars showed lower levels than that of the other 11 broccoli cultivars (Figure 6b).

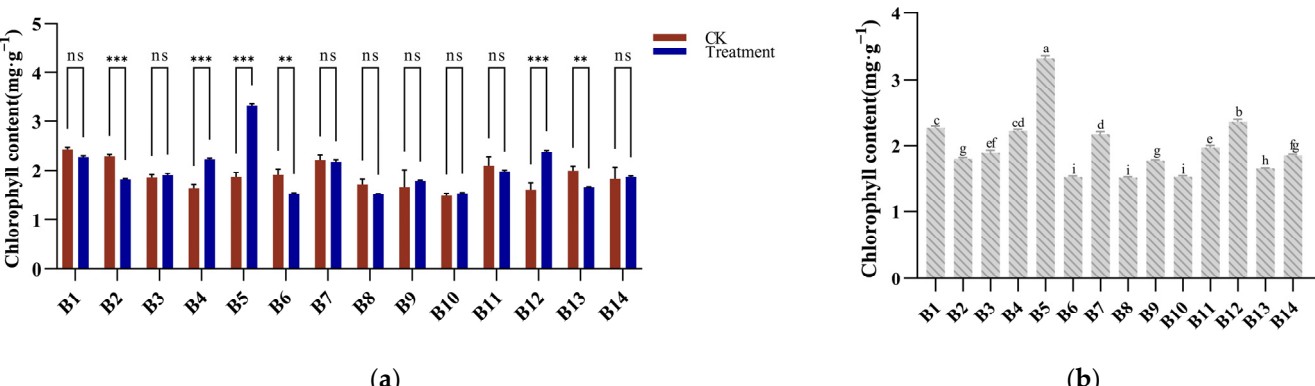

(**a**)                                                                 (**b**)

**Figure 6.** Responses of chlorophyll contents in different broccoli cultivars to cold stress. (**a**) The comparisons of chlorophyll contents among broccoli cultivars before and after low-temperature treatment. CK refers to before low-temperature treatment. Treatment refers to after low-temperature treatment. ** indicates a significant difference at the level of $p < 0.01$, *** indicates a significant difference at the level of $p < 0.001$, and ns indicates an insignificant difference. (**b**) The changes in chlorophyll contents in different broccoli after cold stress using one-way ANOVA analysis. Different lowercase letters indicate significant differences at the level of $p < 0.05$.

### 3.7. Correlation Analysis of CT, EC, MDA, Sugar, and $H_2O_2$

The correlation coefficient matrix between physiological indicators before and after cold stress treatment was obtained using Pearson's correlation analysis (Figure 7). The relationship between total chlorophyll concentration (CT), electrical conductivity (EC), MDA content, soluble sugar (sugar) content, and $H_2O_2$ content was evaluated in this study and shown in Figure 7. There was a significant positive correlation between EC and CT, with a *p* value of 0.007 and a correlation coefficient of 0.687. Meanwhile, there was also a significant positive correlation between EC and soluble sugar content, with a *p* value of 0.033 and a correlation coefficient of 0.571 (Figure 7a).

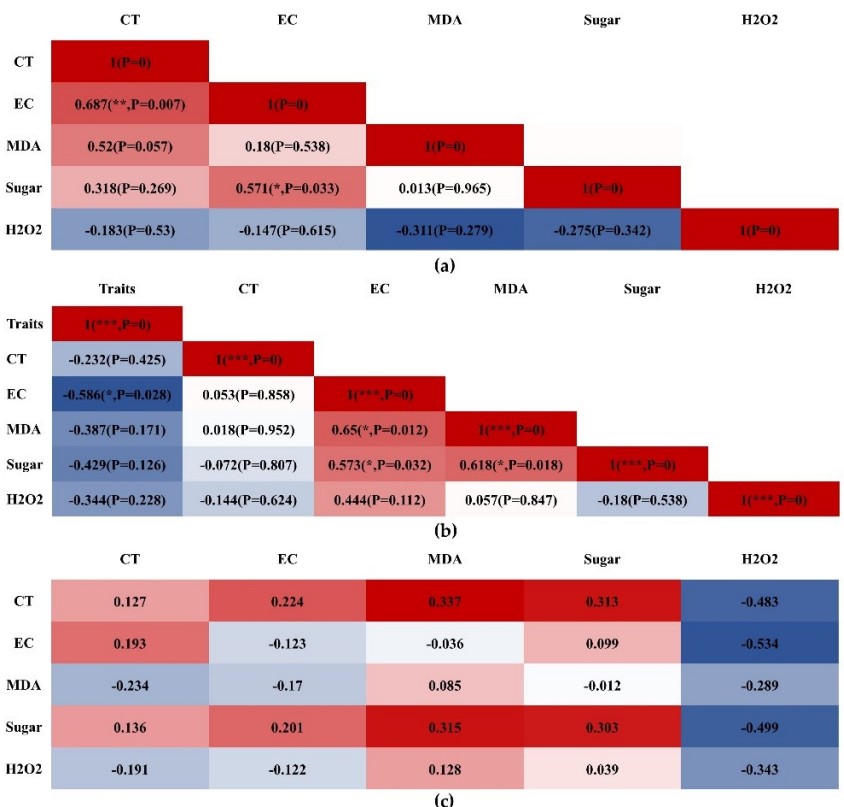

**Figure 7.** Correlation coefficient analysis of CT, EC, MDA, sugar, and $H_2O_2$ before (**a**) and after (**b**), and both (**c**), treatment with cold stress, and *p* value color scale of various physiological indicators. * indicates a significant difference at the level of $p < 0.05$, ** indicates a significant difference at the level of $p < 0.01$, and *** indicates a significant difference at the level of $p < 0.001$.

After low-temperature treatment, the cold tolerance of broccoli cultivars B1–B14 showed a significant negative correlation with EC with a *p* value of 0.028 and a correlation coefficient of −0.586. There was a significant positive correlation between EC and MDA content and sugar content, with *p* values of 0.012 and 0.032, as well as the correlation coefficients of 0.650 and 0.573, respectively. There was also a significant positive correlation between MDA content and sugar content, with a *p* value of 0.018 and a correlation coefficient of 0.618 (Figure 7b).

Pearson correlation analysis of CT, EC, MDA, sugar, and $H_2O_2$ before and after low-temperature treatment, was also carried out to obtain a correlation coefficient matrix (Figure 7c). We could find that there was a negative correlation between electrical conductivity (EC) and $H_2O_2$ content, with a correlation coefficient of −0.534.

In summary, the EC had a good correlation with the phenotype of cold tolerance in fourteen broccoli cultivars, which might be used as the valuable indicator for identifying cold tolerance at seedling stage. Second, the soluble sugar content had a certain correlation with the cold tolerance of broccoli, and it can be verified by the electrical conductivity,

providing a rapid and accurate identification of the cold tolerance in broccoli and the other Brassica plants at the seedling stage.

*3.8. Stomatal Response to Cold Stress*

A comparison was made between the changes in stomatal size and opening status of B13 (Meiqing) with a cold tolerance rating of 5 and B4 (Jade No. 5) with a cold tolerance rating of 0 before and after low-temperature treatment. The results showed similar stomatal size and opening status (Figure 8) with no significant differences, suggesting that stomatal size and opening status do not vary greatly among these different genotypes of broccoli.

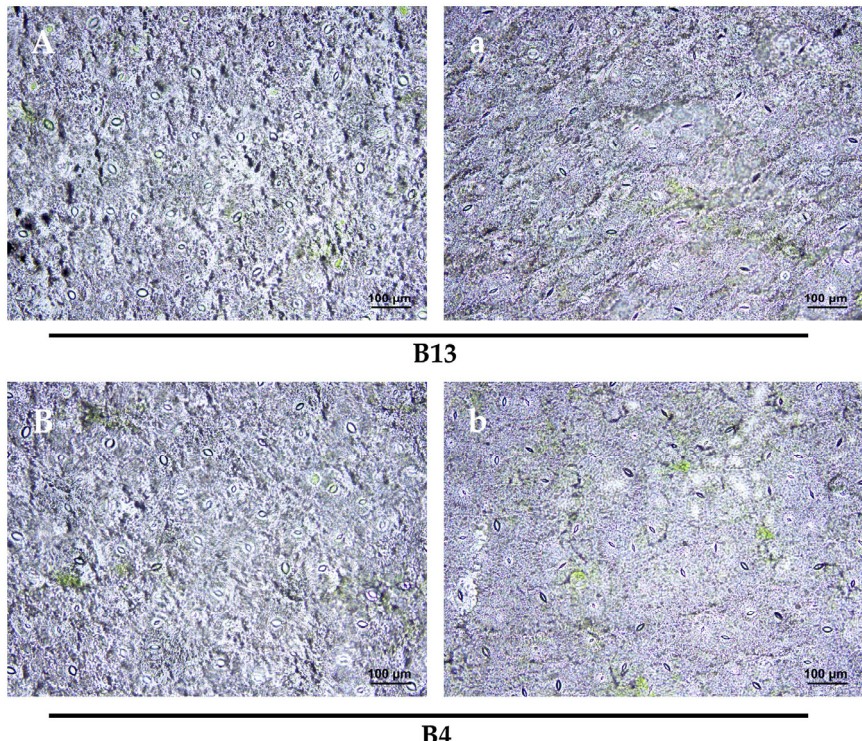

**Figure 8.** (**A**) represents the stomatal status of the lower epidermis of Meiqing leaves before low-temperature treatment, while (**a**) represents the stomatal status of the lower epidermis of Meiqing leaves after low-temperature treatment. (**B**) represents the stomatal status of the lower epidermis of Jade No. 5 leaves before low-temperature treatment, while (**b**) represents the stomatal status of the lower epidermis of Jade No. 5 leaves after low-temperature treatment.

While cold stress is one of the environmental factors affecting stomatal behavior, inhibiting stomatal opening and leading to stomatal closure [19], there are still many other influencing factors, including non-biological factors such as light, $CO_2$, humidity, as well as biological factors like abscisic acid and ethylene [20–24]. Moreover, the process by which these factors influence the stomatal opening and closing mechanism has not been widely studied. Therefore, considering the complexity of the mechanisms influencing stomatal closure, this study does not consider stomatal size and opening status as the identification method for cold tolerance in broccoli.

## 4. Discussion

With the intensification of the influence of climate change on extreme environmental events, along with the frequent occurrence of extreme weather, heat and cold damage has caused a significant reduction in crop yields, increasing the risk to human survival in the future. This calls for strategic planning worldwide, such as the breeding of cold, heat, salt, and drought-resistant cultivars. So it is particularly important to establish a stable and rapid method for identifying the stress tolerance of crops. To date, some rapid identification

methods of stress tolerance have been established for major field crops such as rice and corn during the seedling stage, while cruciferous vegetables are usually identified through field performance, which can be affected by unstable factors including temperature, sunshine, water, insects, and management.

Until now, most studies have focused on the molecular and transcriptional regulatory networks underlying the response of plants to cold stress, such as those conducted on plants like Arabidopsis thaliana, rice, and tomato [25–30]. Research on the cold tolerance of broccoli has mainly focused on post-harvest storage and processing [31–33]. Some studies have identified variables that could serve as biomarkers for cold/hot water stress in broccoli. However, the primary objective of these studies is to produce broccoli that is grown under temperature stress conditions and contains higher levels of beneficial elements for human health [34]. There have been few research reports on the identification of cold tolerance in broccoli seedlings. Here, we present a rapid, stable, and effective identification method.

It is reported that the damage to plants caused by cold stress usually begins in the cell membrane system of the leaves, where chlorophyll synthesis is hindered, causing the increases in biofilm permeability, membrane damage, electrolyte permeability, and leaf relative electrical conductivity [35–37]. Therefore, relative electrical conductivity is generally used as an important indicator to identify the permeability and damage degree of the cell membrane. In our study, after cold stress treatment, the EC of different broccoli cultivars all obviously increased compared to the control at the seedling stage. At the same time, this study proved that there was different response of EC in different broccoli genotypes, providing new insight into cold stress affecting Brassica plants at young stage.

In general, cold stress can induce membrane lipid peroxidation in plants and produce the accumulation of membrane lipid peroxidation, MDA, which is directly proportional to the stress time and inversely proportional to the temperature [38]. In this study, there was a significant increase in MDA in different broccoli genotypes at the seedling stage compared to that in the control after cold stress, which is consistent with some previous research [39].

Soluble sugars are the main components of plant osmotic pressure and highly sensitive to the environmental stresses. This study proved the levels of soluble sugars in different broccoli genotypes except for B7, all increased after cold stress, which is similar to some findings in the ornamental plants, including Chinese cabbage, kale, Brussels sprout, grape, and winter rape, which all indicating the increase in osmotic solutions may enhance plant cold tolerance [40–43].

The findings in *Arabidopsis thaliana* has reported that cold stress can induce the accumulation of $H_2O_2$ in the plant, and reducing the level of $H_2O_2$ can weaken the cold-stress resistance of plants, while increasing the content of $H_2O_2$ will increase the low-temperature response and resistance of plants, indicating that $H_2O_2$ may serve as a signaling molecule for the plant response to the cold stress [44]. In this study, the $H_2O_2$ levels all increased in fourteen broccoli cultivars, except for several cold sensitive genotypes (B1, B2, B5, B6, and B7), indicating that different broccoli genotypes showed horizontal resistance to cold stress which is beneficial to broccoli breeding in cold resistant.

In this study, we proved that some broccoli cultivars, including Meiqing, King 11, and Zhongqing 15, had good cold resistance, which is consistent with the practice of winter cultivars widely planted in Taizhou, Zhejiang province. And we detected that electrical conductivity had a significant negative correlation with the cold tolerance of different broccoli cultivars at the seedling stage. Further research found that the electrical conductivity of broccoli seedlings can be used to identify the cold tolerance in different broccoli genotypes, which can shorten the identification cycle and eliminate the impact of environmental factors. The electrical conductivity and MDA level of cold treated broccoli cultivars both showed a significant positive correlation with the soluble sugar content, indicating that the variability of these three physiological indicators can serve as an indirect standard for identifying the cold tolerance of different broccoli cultivars. Our findings firstly provide a stable, rapid, and effective method of the cold tolerance in different broccoli genotypes, and as well as the other cruciferous vegetable at the seedling stage.

## 5. Conclusions

This study took the lead in constructing a stable and rapid method for cold tolerance identification in different broccoli genotypes at seedling stage, which could save time and avoid the influences of environmental factors to the greatest extent. According to determination of EC, MDA, and soluble sugar values in the seedlings, we might evaluate the cold tolerance of different broccoli genotypes as well as the other cruciferous vegetables. This method has the advantage of good repeatability, stability, and mitigating the shortcomings of long field identification cycles, which provides a scientific basis for identifying cold tolerance and breeding new cultivars of cruciferous vegetables. And we proved that the three broccoli cultivars of Meiqing, King 11, and Zhongqing 15 had good cold tolerance based on this method, which is consistent with the actual planting practice.

**Author Contributions:** Conceptualization, and supervision, Z.L.; methodology, Z.L. and F.H.; analysis, D.W. and Y.Z.; investigation, D.W., Y.Z. and Y.L. (Yumei Liu); data curation, D.W., Y.L. (Yuxiang Liu) and Y.Z.; writing—original draft preparation, D.W.; writing—review and editing, Z.L.; visualization, J.H. All authors have read and agreed to the published version of the manuscript.

**Funding:** This work was supported by the National Nature Science Foundation (32172580), the China Agriculture Research System (CARS-23-A05), the Agricultural Science and Technology Innovation Program (ASTIP).

**Data Availability Statement:** The data presented in this study are available on request from the corresponding author.

**Conflicts of Interest:** The authors declare no conflicts of interest.

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
