# Peer review of "Construction and Identification of Cold Tolerance in Different Broccoli Cultivars at the Seedling Stage"

_agronomy, doi:10.3390/agronomy14020237_

Round 1

Reviewer 1 Report

Comments and Suggestions for Authors

The authors report a descriptive study on the resistance of some broccoli cultivars to abiotic low temperature stress. In particular, the authors used 14 Brassicacea cultivars and exposed them to 4°C for two days, then to 2°C for 16 hours and to -5°C for 8 hours each day for three days. They then returned the plants to 25°C and measured some physiological parameters (electrical conductivity, MDA, sugars, H2O2 and chlorophyll). Finally, they established the three cold-resistant cultivars.

Unfortunately, the authors do not explain this phenomenon. Why were only these varieties resistant? Are there genetic or morphological reasons (cutinization of the epidermis, stomatal distribution)? The manuscript is only descriptive and I believe that more physiological or genetic parameters are needed to explain the phenomenon of cold resistance of the three broccoli varieties.

Reviewer 2 Report

Comments and Suggestions for Authors

Dear Authors, your work sent to me for review is very interesting and contains interesting research results. As a reviewer, however, I must point out some errors. My suggestions for improving the work are included in the comments on the manuscript. These are recommendations that, in my opinion, will improve the work. These are more editorial errors than methodological errors in the research. I believe that the research methods and results obtained and their description presented by you are correct. Also the conclusions are correct. My comments concern the introduction and discussion. The discussion of the results is incomplete and could be further developed and more literature added. Good luck

Reviewer 3 Report

Comments and Suggestions for Authors

Major comment: 

    You write that ‘There is very little research on the comprehensive identification of low temperature tolerance, and plant low temperature tolerance is a comprehensive trait that requires the use of a combination of multiple methods to avoid the one-sided evaluation of plant low-temperature tolerance using only a single indicator of change’-

Why have not you tried to use other indicators, such as plant antioxidant status (antioxidant activity, polyphenol and ascorbic acid content) and proline accumulation? Are these indicators less informative?

Minor comments:

1)     2.7. Determination of chlorophyll content- add extractant and a citation of the method used

2)     Fig.2,6- decipher CK

3)     Fig.7 (a) and (b) the wright upper part of the tables and the left low one completely repeat each other. May be better to delete the repetition?

4)     Why do you indicate your method as a nondestructive- it is not so. May be ‘express method’?

5)     Fig.7 decipher CT,

6)     Change ‘mg/g’ to ‘mg g-1’ Fig.6 (and other Figures, lines 270, 278) and add ‘cutivar’ for the abscissa axis

7)     Fig.6- footnote- can’t understand: there are two ‘b’ indexes ‘The changes of H2O2 levels in different broccoli after cold stress’- there should be only chlorophyll

8)     Citations in the text should not be written in superscript

9)     Figure 7- decipher ‘*’. ‘**’, ‘***’ in footnote

10)  Reference list – revise according to the authors guidelines

Round 2

Reviewer 1 Report

Comments and Suggestions for Authors

In my opinion, the authors did not improve their manuscript as requested. In particular, the evaluation of cold resistance of broccoli seedlings implies that further measurements be made to establish the physiological, molecular, or morphological mechanisms underlying the biological response of some varieties. As mentioned above, could a thickening of the epidermis, different distribution of stomata, or presence of secondary metabolites explain the observed response? We will never know because the authors did not perform measurements in this direction. 

In addition, the screening of cold-tolerant varieties during the planting stage in the main growing areas of China makes the manuscript very descriptive and not very innovative. 

In my opinion, the authors have approached the topic superficially; further analysis is needed.

Comments on the Quality of English Language

Minor editing of English language required

Round 3

Reviewer 1 Report

Comments and Suggestions for Authors

I understand the difficulty and purpose of the authors' work and believe that the manuscript can be considered for publication in Agronomy. However, I believe that the figure of stomata (given in the response to the reviewer) may give more value to the manuscript. I suggest adding it and commenting on it in the main text. After this minor revision the manuscript can be published.
